# Cyber Interpersonal Violence: Adolescent Perspectives and Digital Practices

**DOI:** 10.3390/ijerph21070832

**Published:** 2024-06-26

**Authors:** Bárbara Machado, Paula Lobato de Faria, Isabel Araújo, Sónia Caridade

**Affiliations:** 1NOVA National School of Public Health, NOVA University of Lisboa, 1600-560 Lisboa, Portugal; 2Interdisciplinary Centre for Social Sciences (CICS), Comprehensive Health Research Centre (CHRC), National School of Public Health, NOVA University, 1600-560 Lisboa, Portugal; pa.lobfaria@ensp.unl.pt; 3The Artificial Intelligence and Health Research Unit, Polytechnic University of Health, CESPU, 4760-409 Vila Nova de Famalicão, Portugal; isabel.araujo@ipsn.cespu.pt; 4Psychology Research Centre, School of Psychology, University of Minho, 4710-057 Braga, Portugal; scaridade@psi.uminho.pt

**Keywords:** adolescents, cyber violence, interpersonal violence, digital practices, public health, mental health

## Abstract

Background: The pervasive use of technology, especially among adolescents, has enabled cyber communication and brought many advantages but also led to potential violence. The issue of cyber interpersonal violence (CIV) impacting young individuals is increasingly recognized as a matter of public health; however, little is known about adolescents’ perspectives of the phenomenon. This study explores adolescents’ perspectives on CIV. It seeks to understand their interpretations of abuse, victim impact and reactions, violence escalation, gender issues, victimization and perpetration patterns, and bystander roles. Methods: This qualitative study used fifteen focus groups to gather elementary school participants’ perspectives on cyber interpersonal violence. From four Portuguese schools, 108 participants (*M* = 12.87 and *SD* = 0.31) participated in the study. A thematic analysis uncovered three themes. The results evidenced adolescents’ perspectives about CIV. Due to the amount of time spent online, adolescents regularly encounter cyber harassment and recognize the importance of help-seeking. Mental health problems and their influence on the social and educational lives of adolescents is arising as a CIV problem. Conclusions: Parents play a crucial role in mitigating CIV as well as bystanders. Future programs should promote healthy relationships, raise CIV awareness, involve stakeholders, guide parents, integrate perpetrators into programs, and foster effective networking.

## 1. Introduction

Young people and adolescents are taking the lead in adopting new technologies, as social networks and instant messaging services have become common virtual platforms in our culture. This new generation is named “digital natives” because they were born into and raised in the digital era [1]. The term “digital native” first appeared in the literature in the late 1990s and is credited to Prensky [2,3] and Tapscott [4]. Digital natives are individuals born between 1980 and 1994, representing the first generation to grow up with new technology. They spent most of their lives having access to digital communication technology [5]. Prensky [2] mentions individuals who were not born into the digital world but adapted to it later in life as “digital immigrants”. Later, other authors [5,6] created an extra unifying concept and labelled it as “digital learners”. New social relationships and behaviors are growing and evolving because children and teenagers have access to computers and other electronic devices continually [7]. They cannot picture interaction in a world where such devices do not exist. Adolescence is a long period and can be subdivided into four parts: preadolescence (ages 11–13), early adolescence (ages 13–15), middle adolescence (ages 15–17), and late adolescence (ages 17–19) [8].

Studies [9,10] indicate that around nine out of ten Portuguese adolescents aged 9–17 use mobile phones to access the internet daily, like other European countries [10]. Due to the explosion in the availability of electronic tools among young people, many of their social and communication interaction skills have been lost [11]. These digital behaviors raise the possibility of encountering cyber interpersonal violence (CIV), as more and more people can access global, cross-border digital communication. CIV was first defined as acts intended to damage a victim’s interpersonal connections or self-concept, such as sharing intimate photos of the victim online without permission [10]. Although, adolescents’ social processes benefit significantly from the use of digital tools, these same practices also expose them to more interpersonal intrusion, increasing their risk of experiencing cyber dating abuse (CDA), cyberstalking, cyberbullying, and sexting [12,13]. CDA concentrates on dating relationships and is an emergent form of dating abuse that can be a harmful experience for adolescents, triggering psychological damage in its victims [14,15]. Cyberbullying is defined as the use of electronic devices to intentionally support repeated and hostile behavior by an individual or group, aiming to harm others [16]. Nowadays, with the rise of new technology fields, cyberbullying is defined as suffering of some consideration by researchers [17].

Despite these types of abuse, we understand that adolescents in this generation have access to more educational opportunities, a greater amount of information about entertainment, health, and other topics, and a liberating platform to express themselves, make friends, and form social bonds thanks to communication technologies [18]. However, they also have to deal with the potentially harmful side effects of the digital environment’s evolution, such as unsafe internet usage [19,20]. Despite restrictions for young children on popular social media platforms, their use can still endanger children’s psychological and physical well-being. A study of 14 social media companies’ policies and interviews with their representatives found that cyberbullying continues despite these measures [21]. Some platforms have improved by allowing users to report cyberbullying and requiring real names [22].

Data from several areas show us that CIV has increased. In Korea, different studies revealed that the rates of cyber victimization rose from 19% in 2019 to 19.7% in 2020 [23], with the highest prevalence increasing from 25.8% in 2019 to 32.7% in 2020 among elementary school students [24]. In Australia, from 2015 to 2020 [25], there was an increase in the tendency of cyber victimization over time, with higher rates in 2017 and 2019 compared to 2015; CIV rates in 2020 were also higher than those stated in 2015 and 2019.

In Portugal, a study among adolescents shows that 23% of respondents reported having had a situation that bothered them in the last year, and for 5% this happened at least once a month [26]. Also, the proportion of Portuguese adolescents harassed online increased from 7% in 2000 to 10% in 2014 [27]; by 2022, more than 60% of Portuguese adolescents reported being cyber victims [28]. Managing the adverse effects of CIV calls for a thoughtful theoretical exploration of its roots. Lifestyle-routine activities theory (L-RAT) underlines behavioral and contextual risk factors, contemplates organizational chance as a causal mechanism of CIV victimization, and allows identification of the cyber routines that increase or decrease such risks, directly revealing opportunities for situational prevention and intervention [29]. Although adolescents are born and raised with socio and internet skills, their naïve use of technology, their high risk of exposure to CIV, and a general deficit of coping skills for such experiences make them vulnerable [10,30]. Vale et al. [28] reported that older adolescents who published photographs online, including texts, music, and videos shared with strangers, and who had parents with less knowledge of their online activities were at high risk of being victimized. The sex of adolescents was not considered a predictive characteristic.

Given these potential risks, parents play a crucial role in protecting their adolescents online. They can do this by implementing educational mediation strategies (such as supervision, debating content, co-viewing, and encouragement of solution-oriented communication) and taking a preventive approach (including controls, setting rules, restrictive screen time, and restricted content). These measures not only safeguard children but also foster their self-sufficiency and self-regulation [30]. Portuguese adolescents could benefit from the reassuring support of the Portuguese Safer Internet Centre’s free helpline, with professionals that provide victims support in online crimes, such as information, emotional support, and legal support, as they can report to authorities [29]. Adolescents can find guidance in national awareness campaigns (e.g., Portuguese Association for Victim Support, anti-cyberbullying campaign in schools, SeguraNet) and media campaigns [30]. Finally, adolescents can talk with psychologists in schools and nurses from care centers who visit schools periodically. They can also discuss these matters with teachers, who may provide some training according to their own initiative or, at times, the school’s initiative. L-RAT studies [31,32] found that revealing private information also contributed to becoming a victim. More extensive social networks, resulting in many friends on social media, appear to be a shared risk factor. Connection with strangers can provide access to private information, such as daily routines or physical appearance. Adolescents realize the importance of not talking with strangers as a preventive measure. They also consider parental monitoring of their kids’ activities an essential protective measure [33]. Parents should try to adopt parental control and mediation strategies for their children’s online activities [30]. Considering the technical difficulties that parents can face, prevention programs should integrate internet topics intended for parents.

The meta-analysis conducted by Caridade and Braga [15] emphasized three critical contexts for risk factors of youth CIV: individuals, intimate relationships, and peers. Young people who have experienced victimization and struggle with mental health issues, use alcohol and drugs, or have peers who use and are involved in offline domestic violence are considered a high-risk group. Despite their sociodemographic characteristics, prevention programs could focus on this specific group [15]. This meta-analysis also proposes that the impact of peers as a risk element needs consideration when addressing Cyber Dating Violence (CDV) perpetration. Aligning with social learning processes, it is crucial to foster appropriate and positive social role models within intimate and peer relationships to disrupt the cycle of violence. Additionally, this study revealed that individual sociodemographic factors like gender, age, and ethnicity showed no correlation with youth CDV [15]. Taylor and Xia, in their systematic review, confirmed that both men and women engage in CDV at similar rates, emphasizing the significance of recognizing distinct demographic aspects in prevention strategies [34].

Existing programs have focused on individual and peer risk factors, which, in our view, should remain a priority [35]. It is also crucial to design programs using innovative strategies, such as the bystander approach [36], which has shown promise, for example, in preventing dating violence [37]. This method aims to empower young people to intervene and disrupt potentially abusive situations they witness, with a focus on the influence of peers and other social circles. While raising awareness is vital for preventing CIV, it alone is not enough to address the complexity of this issue. Effective prevention requires sustained efforts that involve a multidimensional and multi-agent approach, incorporating skill-building components [35].

### Present Study

This study aims to analyze perspectives on CIV in a group of Portuguese preadolescents and adolescents. It is also intended to understand the definitions attributed by participants on the subject: meanings attributed to abuse, impact on the victim and reactions, the evolution of violence, gender issues, situations of victimization, situations of perpetration, and the role of bystanders.

CIV deep comprehension is essential for safeguarding young people’s well-being, promoting healthy digital interactions, and developing effective prevention strategies. By understanding the dynamics of CIV, we can work towards creating safer online spaces for everyone. However, there is little research on the meaning that adolescents attribute to such phenomena [38], although this perspective is crucial to developing programs that adjust to their reality. This is why a qualitative approach, with its unique ability to explore the nuances and complexities of the issue, is essential for better mapping the problem among adolescents. It allows us to understand in a more in-depth and holistic way the meanings they attribute to this type of abuse. In Portugal, as in other countries, most studies in this area are based on a quantitative approach [38,39], which does not allow for the deep understanding of CIV that we aim to achieve in this study on national perspectives of young students.

## 2. Materials and Methods

### 2.1. Participants

The research participants were (*N* = 108) 42 boys and 66 girls aged 11–15 (*M* = 12.87 and *SD* = 0.31) from 4 schools across northern regions of Portugal. The selection of schools was based on the following inclusion criteria: adolescents between 6th grade and 9th grade (middle and high school).

### 2.2. Procedures

A qualitative non-experimental investigation was planned. Participation was voluntary, and participants could withdraw from the focus group at any time. To maintain confidentiality and due to the sensitive nature of the research, participants were asked not to use people’s names when discussing incidents involving others and to avoid linking descriptions of behaviors to specific individuals. Participants were free to participate in the focus group from any location they preferred, and they were never directly questioned one by one.

The schools involved were all public. First, the headmaster of each selected school was contacted to determine the school’s interest in participating in the study. If the response was affirmative, a teacher was designated to facilitate all processes between the school and the researchers. This teacher confirmed which students were interested in participating in the study and checked whether their parents agreed. If they did, the researchers contacted the students’ parents/guardians and provided them with the necessary information about the study. After obtaining the parents’ informed consent, the researchers contacted the participants to ask for their consent and scheduled the focus group interview if consent was given. Participants agreed to participate in the focus group and to have it recorded. In their consent forms, parents were informed that the focus group would be recorded and that the data would be used for this investigation.

Information, consent, and recruitment materials were developed and approved within a strict ethical procedure that included approval by the Ethical Committee of the National School of Public Health, NOVA University (number 3/2021), the consent of the Ministry of Education—Monitoring School Surveys (number 0789800001), and informed consent by the parent(s)/guardian(s) of participants. Guarantees were given to preserve and disseminate research materials in a manner that guarded respondents’ confidentiality.

The method of semi-structured interviews was applied in focus groups. The study used a qualitative methodology to explore CIV from the specific contextualized participants’ perspective [40]. Investigating CIV from the perspective of human interactions in a specific place, such as schools, and its cultural context is possible with a qualitative study design [41]. Throughout the semi-structured interview, we pursued covering all the above themes. The script of the interview had macro areas of discussion, such as the definition of CIV and typologies, meanings attributed to abuse, impact on the victim and their reaction, gender issues, contact with aggression, victimization, and bystander situations.

The moderator adopted a neutral attitude to ensure the respondents’ safety and openness. They discussed their experiences with extremely sensitive themes and abstained from judgments and interpretations during the focus group. The leading researcher moderated the focus group.

Focus groups followed the established criteria, such as having only boys or girls from several grades, all picked randomly, to analyze whether there were differences in groups constituted by girls and boys. The rest of the groups included the remaining participants, also distributed in an aleatory way. Focus groups were conducted in May 2022, during afternoons or mornings without teaching activity, in classrooms provided by the schools, in an environment where privacy was guaranteed. The duration of the activity ranged from 45 min to 1 h, depending on the willingness and readiness of the respondents. As we were dealing with preadolescents, it was essential to reduce the risk of inducing negative emotions in participants after asking them to recall episodes of CIV they might have experienced. Therefore, participants were informed that they could be referred to the school psychologist whenever necessary. Additionally, contact information for various support institutions, such as the Portuguese Victim Support Association, was provided and made available.

### 2.3. Data Analysis

The focus group discussion was audio recorded, verbatim transcribed, and verified as accurate. The double review method was used, in which two research team members reviewed each transcript independently. This helped to identify and correct possible transcription errors. To uncover patterns and themes in the data, a thematic analysis was conducted using an inductive methodology, allowing conclusions to be drawn directly from the data, as proposed by Bardin [42]. The researchers completed several readings of the transcripts to create initial codes, which were then refined by more readings, reflexive journaling, mind mapping, and memo writing, to arrange the codes into meaningful groupings to discover overarching themes. Themes and sub-themes were discussed and agreed upon with the research team. During initial and subsequent readings, semantic and open codes were generated until saturation. For data validation, the researchers proceeded to a detailed description, and a co-codification of the data was undertaken. Coding was as inclusive as possible to avoid obscuring any potentially significant extracts in the theme. The NVivo 14.0 software was used to organize, code, and interpret the data. Validation was carried out to ensure the reliability and credibility of the results. Thus, two independent researchers with experience in qualitative data analysis methods and who did not know the participants coded the transcribed interviews. One of the researchers independently coded the entire sample, and the other researcher 20% of the sample (four focal groups interviews), obtaining 89% agreement for the themes. A senior researcher audited the entire coding process.

## 3. Results

In a global analysis, and considering the categories with an excellent representation, five stand out: 1. Cyber Interpersonal Violence (CIV); 2. contacts with cases of victimization; 3. bystanders; 4. risk factors for cyber interpersonal violence; 5. impact of cyber interpersonal violence, and 6. cyber interpersonal violence and gender. Several subcategories emerged from these categories, which will be shown below.

### 3.1. Cyber Interpersonal Violence

Eleven subcategories emerged through the participants’ reports about cyber interpersonal violence (Table 1).

Participants described several forms of cyber interpersonal violence, highlighting three abusive states, namely emotional, physical, and sexual abuse. Participants often said that they had suffered or assisted in all these forms, which can be practiced in isolation or in conjunction with one another (“Many kids insult people via text or email”—P1; “Many times, we saw aggressive and mean comments in chats or on social media”—P2; “Spread rumors about the person”—P3; “Spread fake news”—P4).

Sexual abuse was less common but also mentioned by the participants (“A girl from school had everyone share a nude of her; it was horrible”—P5; “The girls are very mean to each other and comment on each other’s bodies on social media, to assault each other”—P1). Participants also note that “Online threats are often met in person, with physical aggression”—P6.

They also report several abuse typologies, namely cyberbullying (“I think the most common are insults through digital media; they happen all the time these days”—P7; “Many people post photos of others without permission and even make fun of them”—P8); Cyber dating abuse (“Some young people date each other and exchange insulting messages with each other”—P7; “Sometimes girls date boys, share intimate photos with them, and they post them for everyone to see”—P9), and cyberstalking (“A girl was stalked by a boy online. He was always bothering her, and even if she told him to stop, he would not stop. One day after school, he even followed her home”—P10).

Concerning CIV, participants consider that there are some motivations to practice these acts, such as fun (“Some people enjoy making fun of others”—P11), punishment (“Sometimes people are hurt by some past situation and try to get revenge”—P12), discrimination (“Cyberbullying often happens due to discrimination, skin color, way of dressing and even sexual orientation”—P13), and jealously (“They are often jealous of that person and try to put him/her down”—P14). They also consider there are some facilitators to implement this kind of violence, such as digital practices (“Nowadays almost all young people have access to the internet everywhere, both at school and at home”—P1; “We have the internet on mobile phones, so it is always available”—P14; “At school, we all have cell phones or computers. Many also have tablets and other electronic devices where cyber violence can be committed”—P19; “Even in games, with the PlayStation, we can be aggressors or victims”—P20) and individual characteristics (“Young people stay at home more, especially in big cities, so they use digital media more than in the old days when our parents used to play in the street”—P17; “Children have social networks very early on and sometimes they do not even realize the problems they are getting into”—P18; “We all have social networks, and we can post things in real time, things that are often bad and that hurt other people”—P2). Many participants also note that nowadays, the lack of parental control is a violence facilitator (“Parents do not have time and do not control their children’s activities, so they do not know if they are aggressors or victims”—P22; “Young people now have much freedom, they can do whatever they want and there is not much control”—P5; “We have much freedom, we can talk to whomever we want”—P15), associated with the worldwide information spread (“Things are exposed right away, and it is all very fast. And the worst part is that it stays forever, once published there is no way to delete it”—P23).

In this category, a subcategory of abusive behaviors surfaced, where the participants identified various actions. These include lying (“Lying is a serious matter”—P27), disseminating rumors (“Circulating rumors or false information about us is very serious and harmful”—P28), framing others (“Implicating someone in cyber-attacks”—P29), and posting photos or videos without consent (“Uploading other people’s photos without permission is very serious”—P5; “Sharing videos and photos is also serious, not just posting them. If they are explicit, it becomes even more serious”—P11; “I believe that both posting and sharing photos and videos are serious actions”—P30).

Others identified actions like hacking (“Infiltrating social networks, as we discussed in that case, is very serious”—P31), exposing someone’s private life (“Revealing aspects of someone’s private life is serious”—P32), manipulating images (“It’s also very serious when individuals create photo collages and then post them”—P15), making threats (“Threatening the individual or their family is very serious”—P22), hurling insults (“Insults are less severe because it’s easier to ignore the person”—P33), and taking photos without consent (“Taking photos of the person is less serious as long as they aren’t posted”—P34). Participants were also concerned about violence proliferation. They mention recidivism (“Normally, cyber violence does not happen just once, it repeats itself over time”—P35; “The victim is a victim several times because violence is repeated”—P36) and the escalation of violence (“I think it usually starts mild, then over time it gets worse. Sometimes it starts with insults and then they start posting more intimate things about the victim”—P1). Profiles of both the victim and the aggressor were also discussed. The profiles of the aggressor and the victim were often associated with gender (“It’s more common for girls to be victims and boys to be aggressors”—P36; “When the aggressor is a girl, it’s usually girl against girl”—P37; “It’s more common for girls to be victims and boys to be aggressors”—P32). The profile of the aggressor is also linked to social status (“Bullies are not always the popular ones. Sometimes they share characteristics with the victims, and they act this way to feel superior”—P38) and personal traits (“Bullies are not always the popular ones. Sometimes they share characteristics with the victims, like being shy, and they act this way to feel superior”—P38).

Regarding the victim profile, it also included physical characteristics (“People who wear glasses or have a disability are also subjected to mean comments”—P39; “Young people with acne are also made fun of”—P40; “If we are too short, they make fun of our photos on social networks, but if we are too tall, too”—P41), psychological characteristics (“Very shy people are also easy targets because they don’t complain or tell anyone, they usually suffer in silence”—P27), sociodemographic characteristics (“People who are victims are also victims because they reveal on social networks that they are, for example, homosexuals, that they are not Christians and sometimes have ethnicities that are different from ours”—P42; “Being a foreigner can also lead to cyber violence, with mean comments because people don’t express themselves so well in our language”—P43), and social functioning (“The fact that they always take pictures with the same clothes, because they are poor, can already make the person a possible cyber victim”—P11).

Referring to CIV, victim reactions were also cited, such as maintaining a relationship (“Most victims at first do nothing, they try to ignore it and see if the aggressor stops”—P33) and being passive. Some noted acting (“When it was with me, I told my parents, and they helped to resolve the situation”. “I told my mother and my mother spoke to the police”—P7), talking with friends (“Some victims went to their friends”—P44), talking to teachers (“When this happens here at school, we talk to teacher X, and he solves the situation”—P45), confrontation (“The victim should talk to the abuser to try to understand why he/she is doing that”—P4), and leaving the group (“Many times, when it happens in a WhatsApp group, the victim simply leaves the group”—P46). Participants also mentioned they create private accounts (“Have private accounts (…)”—P50), show no fear (“As it is online, it is easier for the victim not to show fear, which can deter the aggressor”—P21), and ask for help (“People should always ask for help, even if it takes time or they think it will not do any good because sometimes it works and problems are solved”—P44). In the matter related to asking for help, other subcategories emerged, including informal help such as talking with an adult (“Talk to parents”. “First, they should talk to a close relative”—P1) or talking with peers (“I told my best friend what was happening so she could find a way to help me”—P42). Some also noted formal help, which includes talking to a professional (“Tell the school psychologist, and he/she might be able to help”—P48), to a police officer (“If it is an extreme case with threats and exposure of photos, I think we should talk to the police”—P5), and to school staff (“Talk to the class director”—P17; “Tell the teachers”—P22; “Tell the operational assistants”—P35).

The last theme that emerged was related to self-protective behaviors, such as do not interact with strangers (“I prefer to play online with people I know because it’s safer”—P47; “Sometimes playing online with people we already know can lead to violence faster, because they know us better and so they know how to attack us with insults for example”—P48), do not share information (“Do not share our passwords with other people”—P49; “Not sharing information about our lives with people we only meet online”—P8), limit the age of access to social networks (“Limit the age of access of children to social networks”—P32), parental control (“Parents must control what their children do online”—P2), parental and school education, respectively (“Parents should talk to their children about cyber violence and teach them how to defend themselves”—P40; “We should hear more about these matters at school”. “The school could promote more lectures on this topic”—P40), limited content (“We share very little of our stuff on social media, for example, we shouldn’t even post pictures of ourselves”—P38), cyber security (“Having an antivirus installed on our computer, so they don’t hack our accounts”—P50), specialized offices (“In schools, create offices with professionals such as psychologists and teachers to deal with these matters”—P17), censured content (“The people who manage the social networks should censor this content, so that it doesn’t spread”—P1), do not take nudes (“Never take nudes, so there is no risk of them being shared”—P43), children’s education (“Educating children not to be mean to other people, teaching them that cyber violence is something very bad and that it can do a lot of harm to someone else’s life”—P3) and existence of a reliable person (“Create a reference person in schools that all participants know they can turn to in these situations”—P52).

### 3.2. Contacts with Cases of Victimization

Participants reported several contacts with cases of victimization in their daily lives (cf. Table 2), which can be divided into types of victimization that include sexual victimization, where the most mentioned kind was “nudes” (“When I was in 5th grade, a girl in my class sent nudes to a boy and he spread them all over the school”—P45), cyberbullying with photo sharing (“This year there was a case in my class with a 7th-grade girl. It all started with anger, then everyone made fun of her and recorded videos and took pictures and then posted them online, in groups, on social networks”—P8), meme creation (“It happened here at school, a group of participants created memes of other participants and even teachers and spread them on Instagram, in the class group”—P1), sexual orientation discrimination (“It was a boy who in a group called him gay. Then they started to put it on his social networks and in other groups he was in and make fun of him”—P53), spreading rumors (“It happened to me, they started writing false information about me on Twitter and Instagram and spreading lies”—P48), hacked accounts (“During the pandemic, a girl in my class reduced the number of guests for her birthday party. A boy, because he was not invited, hacked her Instagram account, and started posting embarrassing things in her name”—P15; “In our class, a person has already hacked another colleague’s email account and sent emails with terrible things to all contacts”—P22), gaming violence (“I have suffered from violence while playing video games. They started insulting me in messages, and then they recorded the game and made fun of me in WhatsApp groups saying I was a fragile player”—P54), and sexual harassment (“A boy from another class made many comments of a sexual character on Messenger to a girl. He even harassed her and even followed her home.”—P36).

In this study, participants only had contact with CIV as victims (“It happened to me, they started writing false information about me on Twitter and Instagram and spreading lies”—P18) or bystanders (“It happened here at school, a group of participants created memes of other participants”—P27). In both cases, they state that a request for help is necessary. That help could be from a professional (“She went to tell the psychologist (…)”—P13; “We told teacher X, and he took care of the situation”—P1; “We went to talk to the school board (…)”—P33), which participants mentioned being a psychologist, a teacher, and an operational assistant, among others; from parents (“I told my mother (…)”—P25; “The father saw the conversation on WhatsApp (…)”—P55); from the police (“My mother went to tell the police what was going on (…)”—P25); or from peer group (“The girl told her friends, who reported it”—P56). However, it was recognized that sometimes there is an absence of requests for help (“Many young people do not ask anyone for help, because they do not trust or are afraid of being mocked”—P41).

### 3.3. Bystanders

Adolescents, in their experiences, reported that they had contact with CIV as bystanders (cf. Table 3). This category includes bystanders’ actions and attitudes. Concerning bystanders’ actions, there is victim support (“We should support victims, so they do not feel alone”; “We must alert the victim to the situation they are experiencing because sometimes they do not even realize that it is cyber violence”—P56), stopping the violence circle (“We must stop sharing what is going around, to stop the cycle of violence”—P57), reporting (“We must report these situations”—P27), talking to the bullies (“We can talk to the abuser to understand why he is doing this”—P58), blocking the bully on social media (“We can help the victim block that person on social media”—P5). Concerning the bystanders’ attitudes, the themes that emerged were ignoring (“I think that many times it is not about liking to see the other being bullied, but about thinking that it is nothing to me”—P7), informal revelation (“We can tell parents, teachers or psychologists about these situations, for them to resolve later”—P40), and encouraging the victim to report (“We must encourage the victim to report the situation”—P52).

### 3.4. Factors Associated with Cyber Interpersonal Violence

According to the participants’ reports, this field has three subcategories (Table 4).

Participants mentioned some factors associated with CIV related to individual characteristics/personality, relational dynamics, and family function. In the subcategory individual characteristics/personality, they stated impulse (“People (aggressors) often do that on impulse, they don’t even think about it, they send a certain message on impulse”—P33), acceptance (“Some do this to be accepted into a certain group of people they consider friends”—P59), anonymity behind a screen (“Feel protected by the anonymity of the digital world”—P60), envy (“They believe that the other person has something they don’t have, it could be a bigger house, a branded outfit and then because they are jealous, they make life hell”—P1), racism/discrimination (“That person has a different skin color or a disability and so that’s why they do it”—P6), mental health issues (“The person (aggressor) has a mental health problem, but is not being treated, such as depression”—P61), for fun (“Some people think it’s fun”—P27), sadness (“The person is very sad about some situation in their life and then takes it out on others”—P44), previous victimization experience (“Some people were once victims and now become perpetrators”—P22; “In childhood they experienced some traumas and hurting others is a way of dealing with their own suffering”—P26), low self-esteem (“They have low self-esteem issues, so they try to hurt others to make themselves feel better”—P15), and child education (“He/she was educated that there was no problem with violence and that is why he/she is violent”—P33). In the second subcategory, relational dynamics, participants included jealousy (“They are jealous of that person for some reason and try (abusers) to make that person feel bad”—P28) and revenge (“That person has already done something to him/her, and so he/she is taking revenge for that situation”—P62). Participants also considered family function, namely divorce (“The family is going through a divorce process; the person is not able to deal with the situation and takes it out on others”—P63), family violence (“The father beats the mother or vice versa”—P2), mourning (“Someone in the family dies and they are unable to grieve and then they become aggressors in order to feel somehow better”—P41), and economic issues (“They are having financial problems at home and try to divert their attention by taking it out on others”—P64).

### 3.5. Adolescent Perceptions about Impact of Cyber Interpersonal Violence

Concerning the impact of CIV (cf. Table 5), participants mentioned two subcategories: emotional changes and behavior changes. In the emotional changes subcategory, they highlighted sadness (“They feel very sad”—P8; “I was very sad about the situation”—P10), shame (“The victims are ashamed of this situation”. “Victims feel ashamed and do not report”—P58), fear (“Victims are afraid of the aggressor”—P44; “They are afraid to expose the situation”—P45), acceptance (“Sometimes they just accept the situation, even if they don’t understand it”—P41), anger (“Victims can also become very angry about the situation”—P60), insecurity (“Victims feel insecure and anxious”—P1; “They feel very insecure”—P2; “Victims are very suspicious and not able to trust anyone”—P25), inferiority (“They feel inferior to the bully and inferior to everyone else”—P65), depression (“Victims feel very sad and may become depressed”—P54), loneliness (“They feel very lonely”—P1; “They feel very alone”—P2), self-esteem problems (“They think something is wrong with them”—P59; “They (victims) think they are the problem”—P57), humiliation (“The victims feel humiliated by the aggressors”—P21), and anxiety (“Victims feel insecure and anxious”—P43). CIV can lead to behavior changes, such as self-harm (“They feel like disappearing and can hurt themselves”—P8), suicide (“It can lead them (victims) to commit suicide”—P1; “(victims) Can commit suicide”—P2), cyber revenge (“If the abuser is very mean and the victim has had enough, the victim may want to get back at him”—P65), social isolation (“The victim isolates himself, starts to walk alone”—P66; “They do not want to leave the house, not even to do activities that they used to enjoy very much”—P67), and eating disorders (“They lose their appetite”—P62; “Some (victims) begin to eat very little”—P61).

### 3.6. Cyber Interpersonal Abuse and Gender

The topic of cyber interpersonal abuse and gender (cf. Table 6) encompassed three subcategories: gender roles, societal values, and the impact of Cyber Interpersonal Violence (CIV). In terms of gender roles, participants pointed out stereotypical roles for females and males, respectively (“Girls are expected to play with dolls, failing which they may be subjected to ridicule”—P55; “Boys are expected to be adept at sports, like basketball, or else they can be ridiculed”—P56).

As for societal values, they were discussed in the context of both males and females (“Boys also face societal beauty standards such as being tall, muscular, blond, and blue-eyed”—P58; “Girls face significant pressure from beauty standards, which include being thin, tall, blonde, and smart, and not having acne. The societal pressure for girls to conform to these standards is particularly intense”—P59).

Finally, participants believed that the impact of CIV varies between males and females (“Boys, I believe, do not suffer as much because they tend not to be as affected by insults or pejorative jokes, especially those related to appearance”—P3; “Girls are greatly affected by cyberbullying, particularly when it pertains to appearance. Moreover, they can be very harsh towards each other, resorting to physical insults”—P68).

## 4. Discussion

The present study finds numerous considerations about CIV among adolescents, mainly among Portuguese ones. It provides an in-depth view of adolescent’s perceptions of CIV, the meaning they attribute to abuse, impact on the victim (e.g., mental health issues, isolation, eating disorders), victim reaction (e.g., ignoring, asking for help), the evolution of the violence cycle, gender issues (e.g., girls tend to suffer more than boys as a result of image manipulation on social media), the situation of victimization (e.g., nudes, gaming violence), characteristics (e.g., the degree to which the aggressor and victims overlap), reaction, and help-seeking. Participants also reflect on their perception as bystanders and their role in CIV cycle disruption (e.g., victim support, reporting). Our study allows a deeper understanding of CIV by providing an in-depth approach to the phenomenon. It examines not only the types of abuse but also the context in which this abuse occurs, the associated factors, the impact, and even gender issues. This comprehensive understanding is crucial for developing effective preventative measures and interventions tailored to address the specific needs and circumstances identified in the research.

Adolescents live in a digital age, and as a result, the growth of digital connections through devices and human interactions has given rise to new issues, including CIV. Increased internet usage suggests a comparatively high perceived advantage from the activity, with both increased frequency and breadth, which would raise exposure to CIV contacts. A higher internet usage rate may also lessen the need for self-defense techniques like changing digital behaviors to thwart CIV. The possibility that a particular victim would seek professional help varies based on the circumstances surrounding the cyber abuse, potential repercussions, and individual coping strategies.

### 4.1. Typologies of CIV

Participants identified several abusive typologies of CIV, considered emotional, sexual, and psychological abuse, reflected in cyberbullying, CDA, and cyberstalking. Several studies [9,10,11,12,41,43] have examined and deepened the data on the different forms of CIV identified herein by students. CDA is a topic heavily studied, with many studies in the field aiming to understand myths and sexting practices [44] as well as conceptualizations and prevalence rates [45,46]. Strickland et al. [47], in their study, suggested that parent–child closeness moderates the harmful effects of CDA on mental health. Also, emotionally close parent–child relations protect against depression and anxiety. Participants mentioned the role of parent supervision in combatting CIV and parents’ role in their child’s support.

Cyberbullying is a subject of high concern nowadays. As mentioned by the participants, cyberbullying can include posting photos and videos from others without consent and spreading rumors on social media, among others. Prevalence rates [48,49] have been studied, as well as predisposing factors [48], and their impact on adolescent life as an adverse childhood event [50,51]. Participants mentioned the effect of cyberbullying on trust and talked about the fear and shame of seeing their lives exposed to everyone.

Cyberstalking is also a growing problem, which the participants identified. Nowadays, its prevalence rate is almost 14% of the total CIV actions, and most of the victims (82.1%) are victimized by someone who is not a friend or a former romantic partner. When asked about the duration of their experiences, 82.1% stated that their torment lasted for six months or less, and for most of them (67.9%), the incident occurred over a year ago. Among the victims, 96.4% encountered a behavior involving communication technology, while 92.9% experienced a behavior that took place on social media [52]. These results seem to agree with ours, as many participants reported being a victim at least once.

### 4.2. Facilitators of CIV

Adolescents indicated digital practices, individual characteristics, and lack of parental supervision as facilitators of CIV. The last two are more associated with cyber dating violence and cyberbullying. They also mentioned that recidivism and escalation of violence as attributes present in all the types of CIV identified. Mishra et al. [53] suggest that with the rise in digital crimes, young people might need to realize their social media actions’ wide-ranging effects in real life. Our study demonstrates that adolescents could identify the main categories of CIV presented in the literature, appearing aware of the risks. Also, their input made it possible to understand their knowledge about the most common types of CIV, namely cyberbullying, defined as the use of digital technology and various electronic devices, including computers, tablets, and cell phones, to harass, threaten, and injure victims [54]. In our study, adolescents also mentioned facilitators of abuse, such as digital practices, which demonstrates their understanding of this kind of abuse. They extended this concern to all kinds of CIV.

Cyber dating abuse was identified by the participants as a form of CIV. The participants identified cyber dating abuse as a form of CIV. Cyber dating abuse is defined as actions motivated by attachment anxiety, clinginess, wrath, jealousy, and other negative emotions that are used to control, monitor, compel, harass, and stalk a romantic partner using digital technology and engage in online partner monitoring [43,55,56]. Participants refer to motivation for CIV as fun, punishment, discrimination, and jealousy. They argue that sometimes adolescents practice CIV because they enjoy making fun of others. Participants also mentioned that sometimes they felt jealous of each other and practiced cyber violence mainly as an impulse. They also specified that sometimes they felt jealous of friends and tried to make fun of that person, but they said it was widespread for people to feel envious of a boyfriend or girlfriend and stalk them on social media or try to control their social accounts. Sometimes, boys made fun of a photo their girlfriend posted because he was jealous and wanted to make her feel bad, and then it was removed. Participants also stated that girls sometimes controlled boyfriends’ messages and likes or comments from other girls on social accounts. Participants identified such behaviors as CIV, which is also referred to in the literature, where studies confirm that cyber dating behaviors include but are not limited to sending threatening messages, stalking partners on the internet, and various forms of surveillance, like checking in on a partner’s account frequently, requesting passwords for phones or online accounts, and keeping an eye on their online activities [57,58]. Despite adolescents’ perceptions about the phenomenon, this subject should continue to be investigated and included in educational programs during adolescence, because studies show very different prevalence rates, ranging from 22% to 50% [12,59].

### 4.3. Abusive Behaviors

Participants also identified the main abusive behaviors presented in CIV, such as telling lies, spreading rumors, framing others for something that they did, and sharing photos online from friends, colleagues, partners, and ex-partners. They also identified the sharing of videos and nudes, hacking others’ email and social media accounts, showing someone’s private life on social media, threatening and insulting others through a digital platform, and taking pictures of someone without consent. Washington [60] aligns with this perspective, saying that instances of cyberbullying encompass a wide range of online behaviors, including name-calling, cyber-harassment, cyberstalking, online impersonation, coercive text messaging, online rumors and gossip, insertion of offensive remarks and comments, sharing or disclosing passwords without authorization, removing someone from a chatroom, posting offensive or embarrassing messages or photos about someone on a public forum, manipulating images, text, videos, and polls to deceive others, and transmitting harmful information that can have disastrous real-world effects. Participants report that the cyberbullying typology was sometimes an extension of bullying or evolved from cyberbullying to bullying when adolescents frequented, for example, the same school. Additionally, there is proof that younger people are more likely than older people to be involved in cyberbullying, with many of them reporting acting as bullies, victims, or both in a range of online spaces such as chat rooms, instant messaging services, and social networking sites [61]. The online space enables anonymity, giving the aggressor a powerful tool to practice CIV. Besides the anonymity, participants also reported as violence facilitators individual characteristics such as being shy, spending more time at home using social and electronic devices, and having access to social media and electronic devices in an early stage of their development. These could lead to the emergence of CIV since, according to L-RAT, when a motivated perpetrator finds an attractive target, several hours spent online by adolescents can exponentiate their exposure [31,48].

They also mentioned some concerns about the lack of parental control in adolescents’ activity, saying that parents do not have time to control their children’s activities, so they fail to identify at an early stage if their children are victims or aggressors. Several factors, including environmental factors (e.g., exposure to media violence, peer pressure), personal factors (e.g., socialization, family upbringing), and personality traits (e.g., moral disengagement, impulsivity, narcissism), have been noted in previous studies to explain the increasing incidence of cyberbullying [62,63]. However, the intensity of cyberbullying has been connected to aggressive behavior, drug abuse, and delinquency [64]. It has also been linked to several mental health issues, such as psychosocial maladjustment, psychiatric disorders, antisocial behaviors, suicidal thoughts, self-harm, anxiety, depression, and a lower quality of life [65,66,67]. Our results corroborate these findings.

### 4.4. Gender Issues

Concerning gender issues, adolescent cyberbullying and cyber dating violence were the CIV issues most often mentioned as gender-determined. Most respondents said that girls were most likely to be the victims and boys the aggressors. They also referred to sociodemographic factors, such as ethnicity and sexual orientation, leading to becoming victims, with physical characteristics outside the norm potentially leading to abuse, such as being overweight. In addition to gender issues, participants also mentioned family issues as a risk factor for CIV (e.g., a troubled divorce of their parents, growing up in a house with family violence, dealing with mourning, or economic issues) in adolescence; participants mentioned this as leading people to become victims or perpetrators of CIV.

Most adolescents consider CIV a vital subject because of its impact on victims. The participants referred to emotional and behavioral changes, which include self-esteem problems, mental health issues, anxiety disorders, eating disorders, and cyber revenge. Referring to the last topic, they believed that, especially in cyberbullying, it is straightforward for the victim to become an aggressor and vice versa.

### 4.5. Contacts with Cases of Victimization

When questioned about their contacts with cases of victimization, most of the participants identified several types of victimization, such as nudes, memes, violence through gaming in real time, sexual harassment, hacking accounts, photo sharing, and discrimination based on sexual orientation. Most of the participants knew or were in contact with cases of sexting and revenge porn; most of them were bystanders, though some of the participants were victims of this kind of CIV. The act of sending a pornographic or semi-pornographic picture, video, or text message using a computer, messaging app, or mobile device to flirt, plead, or express desire is known as sexting [68]. Many of the victims said that they started exchanging messages with someone they knew, and he asked to send sexual pictures after a few years of conversation and mutual interest. The literature states that sexting is frequently used as a lead-up to actual sexual activity, as it can be seen as an extension of concurrent sexual contact to promote bonding and communication attraction [54]. This characteristic suggests that victims need assistance in fully understanding the risks of sending online sexual content. Our participants related to this concerning reality because a considerable number of them admitted to having received or sent semi-nude pictures by message, a reality also reported in another study [69]. Sexting can generate feelings in adolescents such as amusement, enthusiasm, vulnerability, humiliation, and distress. Many of them recognize that when this type of picture comes to public attention, adolescents have a terrible time; they move from one school to another school to avoid colleagues that have seen the picture, they feel very ashamed, sad, anxious, and exposed, and some also move to another city. Participants are aware that this type of publication leads, many times, to children committing suicide or having mental health problems, such as depression. Adolescents recognize sexting and revenge porn as two of the most impactful forms of CIV due to their emotional changes in victims. Most of them were reminded of the victims they knew when they mentioned the impact of CIV, namely emotional changes (e.g., fear, insecurity, self-esteem problems) and behavior changes (e.g., suicide, social isolation, eating disorders).

### 4.6. Bystanders’ Role in CIV

Adolescents believe that the role of bystanders is to support the victim and report or talk to the aggressor in order to stop the violence circle. However, many respondents also recognized that bystanders often ignore these situations because they do not believe they have anything to do with them and do not want to get involved.

A recent systematic review found that in 100% of the studies, the role of a cyber-victim was identified; in 39%, the role of a cyber-aggressor; in 16%, the role of a cyber-observer [70].

### 4.7. Request for Help

Adolescents request formal and informal help, and help-seeking is considered one of the most common coping strategies among victims [38]. However, a smaller number of cyber victims seek help when compared to victims in real space [71], even though most of them understand it is a helpful strategy in controlling the impact of CIV and stopping the violence cycle [72], as recognized by participants. They seek assistance from professionals such as psychologists, teachers, school boards, and operational assistants. They also mentioned contacting police officers. On the other hand, they seek informal help from parents and peer groups. At least some adolescents recognize that sometimes they do not seek help because they are being mocked. These data align with the literature, which says less than 30% of people who experienced technology-facilitated abuse sought professional help [73]. Our results are a little bit different from other results, as participants mentioned that adolescents request informal (e.g., parents, friends) and formal help (e.g., teachers, police officers), without any preference [74,75]. These results could be because many of the participants mentioned being bystanders. As a result, they recognize the importance of seeking help from authorities, for example, due to the effectiveness of this in stopping CIV.

In this line, participants refer to victims’ reactions to CIV, and most of them are described as passive (e.g., maintaining the relationship or simply doing nothing). In contrast, others are active (e.g., talking with friends, leaving the aggressive group, showing no fear, asking for help).

### 4.8. Self-Protective Behaviors

Participants also mentioned some self-protective behaviors that could be implemented to reduce CIV. They identified the need to hear more about the subject at school. A study showed [76] that most participants discussed the value of a continuous education campaign and raising awareness about CIV as a proper preventative strategy that should integrate and continue to educate kids in all grades. In the same study, a school counsellor expressed similar views regarding the effect of education on this subject and how educational programming has raised awareness and communication regarding CIV. He also mentioned that some school programs have been beneficial in getting things out in the open and getting people to talk about cyberbullying, for example. It was also noted that those who engage in CIV should be aware of the legal consequences, which are poorly addressed in school programs [76]. The following are also considered necessary protective measures: do not interact with strangers and instead limit interaction to friends; do not share information such as passwords, and do not take and share nudes. Finally, many participants reported the need for parental control as a protective measure as well as parental education. They believe that parents should control their kids’ digital practices and limit access to inappropriate content. In the current literature, parental involvement with technology is noted as crucial in preventing CIV. Parents are an essential resource for minimizing and resolving youth CIV issues, namely because adolescents are most likely to talk about parental technology use, particularly as a tactic to stop or step in when cyberbullying occurs [76]. Participants also note that parents are responsible for children’s education and should teach them to be kind to one another and recognize several forms of violence to protect themselves against. In the end, participants talked about the possibility of creating specialized offices in school with a multidisciplinary team that is able to support them in the case of CIV, as victims, bystanders, or even aggressors. This office could also include a point person to whom adolescents could go for support. We also verified that some schools already have this person, even in an informal form.

Fortunately, other stakeholders in the literature appear ready to embrace such educational endeavors in their schools. Gradinger et al. [77] examined the opinions of teachers and parents about cyberbullying prevention and discovered that 95% of parents and 90% of teachers have positive views regarding facilitating and participating in anti-bully education strategies. Consequently, teachers and parents would likely be supportive and invested in implementing such educational programming [77]. Other research has documented that teachers and educational support professionals have indicated a desire to receive additional training related to CIV interventions [78]. Additionally, a study investigated the effectiveness of such programs and acknowledged the value of implementing such educational programs [79]. The results of this study speak to those suggestions, as CIV awareness, digital citizenship training, empathy, respect, and conflict management exercises can be easily implemented as new modules within existing programs on the specific matters of CIV, such as cyberbullying and cyber dating violence. At last, Yurdakul and Ayhan’s study [80] concluded that the Cyberbullying Awareness Program successfully enhanced the understanding and coping abilities of teenagers in the intervention group towards cyberbullying. Based on these findings, recommendations are put forth for educators and policymakers, as mentioned above by other authors. Policymakers are advised to incorporate prevention programs with regard to cyberbullying into national curriculums. This would help to raise teenagers’ awareness about cyberbullying and strengthen their resilience. Educators should also deliver these programs to children and teenagers nationwide.

## 5. Practical Implications

Participants identified emotional, physical, and sexual abuse related to CDA, cyberbullying, and cyberstalking. Adolescent digital practices, jealousy, and a lack of parental control contribute to the spread of CIV. Posting, sharing photos and videos, and telling lies are some of the abusive behaviors that adolescents practice or suffer from. Victims should seek help, especially formal help, and the bystander’s role in stopping the cycle of violence should be recognized. Parental control, children’s education, and safe digital practices are some protective measures to be considered. CIV has a significant impact on adolescents’ mental health, with emotional changes (e.g., sadness, fear, anger) that can lead to social isolation or even suicide. Considering this, our research emphasizes the need for prevention through education. Research has highlighted that less than half of the programs developed to combat CIV incorporate educational content for parents. However, such programs are among the most successful at reducing cyber victimization [81]. So, schools must have a clear objective and policy to guard against the many cyber threats that young people encounter nowadays. Schools must also have a program integrating CIV as a multidimensional and multidisciplinary issue directed at parents, victims, aggressors, and bystanders. Researchers should explore bystanders’ roles in stopping CIV, as they can be allies in the fight against CIV, often as informal supports to whom victims and even aggressors turn.

Health professionals, such as psychologists, school nurses, and others, must support and counsel adolescents, as the most reported as the formal help to whom victims and even bystanders turn when looking for advice. Taking into consideration the evidence-based support provided by this study, we suggest that future educational and intervention programs emphasize the following: (1) promotion of healthy relationships; (2) disclosure of media campaigns; (3) awareness about the nature and consequences of CIV; (4) the inclusion of parents, teachers, and other stakeholders; (5) providing guidance and informatic assistance to parents; (6) integrating aggressors in education and restructuring programs; and (7) effective networking (e.g., families, education, legal, health).

## 6. Limitations and Future Research

This study has limitations, although its findings have substantial implications leading to several important recommendations. This study brings a Portuguese perspective provided by adolescents on the CIV topic. However, the findings of this study are based on a sample from a limited geographical area (the northern region of the country). Therefore, samples of adolescents in other parts of the country should be also investigated to understand this phenomenon over a wider horizon. Additionally, the sample size is limited and thus only partially representative of some adolescents in schools; for example, we did not interview participants from private schools. Furthermore, the leading researcher also moderated the interviews. In future studies, an additional observer could be present to better capture and analyze participants’ behaviors. One future direction could be to use the results of this study to create a survey to check these conclusions with a broader sample. Future research should continue to ask adolescents and other stakeholders (e.g., police officers, teachers, and family support professionals) about other types of potential school support available in their geographic area, as the responses may provide different suggestions for solutions.

Future research may also include the victim’s perspective as well as that of the offenders.

## 7. Conclusions

This study complements the current literature by providing evidence about adolescents’ perspectives on CIV. Furthermore, one of the most relevant contributions of this study about CIV is the perspective of the phenomenon from the view of subjects who live this reality in the first person. Moreover, this study demonstrates that the role of parents is increasingly crucial in protecting and educating their children through supervision and re-education regarding their digital practices. This study also indicates that there are specific characteristics, such as family issues and gender issues, specific to victims and aggressors and that there are variations in the severity and the episodes of CIV. These factors can be used to recognize more susceptible adolescents and aggressors, who pose a greater risk of serious CIV, resulting in damage to victims. Finally, this study integrates the bystander’s position on intervening in CIV as a mediator, intervener, or stopper of the cycle of violence.

## Figures and Tables

**Table 1 ijerph-21-00832-t001:** Systems of categories, subcategories, and specific categories regarding cyber interpersonal violence.

Categories	Subcategories	Specific Categories
**Cyber Interpersonal** **violence**	**Abusive forms**	Emotional abuse
		Sexual abuse
		Physical abuse
	**Abuse typologies**	Cyberbullying
		Cyber dating abuse
		Cyberstalking
	**Motivation**	Fun
		Punishment
		Discrimination
		Jealously
	**Violence facilitators**	Digital practices
		Individual characteristics
		Lack of parental control/supervision
	**Abusive behaviors**	Telling lies
		Spreading rumors
		Framing
		Posting and sharing photos, videos, and nudes
		Hacking
		Show off private life
		Image manipulation
		Threats
		Insult
		Take photos
	**Violence proliferation**	Recidivism
		Escalation of violence
	**Aggressor profile**	Gender (male)
		Social status
		Personal characteristics
	**Victim profile**	Sociodemographic characteristics (gender, sexual orientation, ethnicity)
		Physical characteristics
		Psychological characteristics
		Social functioning
	**Victim reactions**	Maintain relationship
		Talking with friends
		Being passive
		Confrontation
		Leaving the group
		Private accounts
		Show no fear!
		Ask for help
	**Help typologies**	Informal help (peers, family…)
		Formal help (police, professors, health professionals…)
	**Self-protective behaviors**	Do not interact with strangers
		Do not share information
		Limit age of access to social networks
		Parental control
		Parental education
		School education
		Limited content
		Cyber security
		Specialized offices
		Censured content
		Do not take nudes
		Children’s education
		Reliable person

**Table 2 ijerph-21-00832-t002:** Categories, subcategories, and specific categories regarding contacts with cases of victimization.

Categories	Subcategories	Specific Categories
**Contacts with cases of** **victimization**	**Types of victimization**	Sexual (Nudes)
		Cyberbullying (photo sharing)
		Memes
		Sexual orientation discrimination
		Spreading rumors
		Hacked accounts
		Gaming violence
		Sexual harassment
	**Role**	Victim
		Bystander
	**Request for help (formal and informal)**	Professionals
		Parents
		Police
		Peer group
		Absence of request for help

**Table 3 ijerph-21-00832-t003:** Categories, subcategories, and specific categories regarding bystanders.

Categories	Subcategories	Specific Categories
**Bystanders**	**Actions**	Victim support
		Stop the violence circle
		Report
		Talk to the bullies
		Block the bully on social media
	**Attitudes**	Ignoring
		Informal revelation
		Encourage the victim to report

**Table 4 ijerph-21-00832-t004:** Categories, subcategories, and specific categories regarding factors associated with CIV.

Categories	Subcategories	Specific Categories
**Factors** **associated with CIV**	**Individual characteristics/personality**	Impulse
		Acceptance
		Anonymity behind a screen
		Envy
		Racism/discrimination
		Mental health issues
		For fun
		Sadness
		Previous victimization experience
		Low self-esteem issues
		Child education
	**Relational dynamics**	Jealousy
		Revenge
	**Family function**	Divorce
		Family Violence
		Mourning
		Economic issues

**Table 5 ijerph-21-00832-t005:** Categories, subcategories, and specific categories regarding impact of cyber interpersonal violence.

Categories	Subcategories	Specific Categories
**Impact of CIV**	**Emotional changes**	Sadness
		Shame
		Fear
		Acceptance
		Anger
		Insecurity
		Inferiority
		Depression
		Loneliness
		Self-esteem problems
		Humiliation
		Anxiety
	**Behavior changes**	Self-harm
		Suicide
		Cyber revenge
		Social isolation
		Eating disorders

**Table 6 ijerph-21-00832-t006:** Categories, subcategories, and specific categories regarding interpersonal abuse and gender.

Categories	Subcategories	Specific Categories
**Interpersonal abuse and gender**	**Gender roles**	Stereotypical female roles
		Stereotypical male roles
	**Social values**	Male
		Female
	**Impact of CIV**	Male
		Female

## Data Availability

The datasets presented in this article are not readily available because this project remains in progress, and data analysis is ongoing. Requests to access the datasets should be directed to bpi.machado@ensp.unl.pt.

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
