# Peer review of "Cyber Interpersonal Violence: Adolescent Perspectives and Digital Practices"

_ijerph, 2024, doi:10.3390/ijerph21070832_

Round 1

Reviewer 1 Report

Comments and Suggestions for Authors

Thanks for the opportunity to review the paper Cyber interpersonal violence: Adolescent Perspectives and Digital Practices. I enjoyed this piece and highly appreciated how the author addressed the gap in knowledge in this research literature. Also, I reckon that as the paper will be of interest for the practice of caregivers, school psychologists and educators, the manuscript should be published after some minor revisions.

I have a few suggestions, as I documented below, to further strengthen argumentations and to streamline the text:

Introduction

Although your sample fall into the umbrella term ‘adolescents’, I would spend some word on this phase of life, in order to specify that you are going to assess preadolescence (ages 11–13) and early adolescence (ages 13–15) (e.g., Colarusso, 1992);

Also, I would spend some word on the differences between cyber interpersonal violence and cyber dating violence – and the difference from cyberbullying -, that seems to be – in this version of the manuscript – treated as the same phenomena;

Within the literature review, the author(s) might want to better justify their hypothesis by commenting on very recent publications on the current prevalence rates of cyberbullying and cybervictimization: i.e.,

Sorrentino, A., Sulla, F., Santamato, M., di Furia, M., Toto, G. A., & Monacis, L. (2023). Has the COVID-19 Pandemic Affected Cyberbullying and Cybervictimization Prevalence among Children and Adolescents? A Systematic Review. International Journal of Environmental Research and Public Health20(10), 5825. https://doi.org/10.3390/ijerph20105825

Also, the results of this systematic investigation question the link between increased internet use and increased of cyberbullying/victimization (making what you write on lines 42-44 open to question).

Moreover, I suggest to give more information (for the benefit of non-Portuguese reader) about existing policies about CIV/Cyberbullying/tradition bullying/aggression; e.g., is there in your schools a teacher whose deputy principal/referent/contact person for bullying issues? Do you have school psychologists in your schools? What training have teachers in public school about aggressive behaviours in school? Is it mandatory? What ‘training’ have kids? – reading the results, your participants sound quite aware of the phenomenon; plus, they trust their teachers as potential advocates/defenders; in other European countries this is not necessary the case, probably because of the lack of training to teachers (?)

Present study

Line 128, I would change ‘This study aims to analyze adolescents' perspectives on CIV’ into ‘This study aims to analyse perspectives on CIV in a group of Portuguese pre adolescents and adolescents’, as your sample is not representative – which is completely fine for a qualitative study

Materials and methods

Participants

Are the schools involved private or public schools?

Procedures

I suggest to provide further information on the procedure:

Did you provide the participants with a definition of CIV before or during the focus groups? In general, seeing the structure/script of the interview would be a good thing, for the sake of replicability – lines 156-157, you say ‘we pursued to cover all the above themes’, but I’m not sure what’s the above themes; I believe are the ones you mention in the results, but you’ve never mentioned that before.

Where were the focus groups conducted? In participants’ school? During school hours? What were the characteristics of the room where you conducted them (e.g., privacy)?  

From who they were conducted?  Did you have a conductor and an observer per each focus group? Was the conductor a psychologist?

How did you reduce the risk of inducing negative emotions in children after asking them to remind about episodes of CIV they might have experienced? Even if the conductor was not a psychologist, was a psychologist available for debriefing in case of induction of negative emotions during the interview?

What was the modality of the focus group? For example, did you ask every participant to reply to each of the foreseen question of let them free to reply if they wanted? Please, provide more info on this.

Data analysis

Lines 188-189, the authors might want to explain how did they verified that the transcription was accurate.

Within this paragraph you speak about researcher and research team – I suggest being more specific: e.g., was the researcher the same person that conducted the focus group? How many people there were in the research team that analysed the transcriptions? (sometimes you speak about the authors instead of researchers; if the research team and the authors line of this manuscript are the same people, you should be fine with just writing the authors, but, please, I suggest to be consistent in how you refer to the people that did the data analysis)

I believe a reference is needed for the NVivo software

Results

I don’t get why you put cyber in brackets. That was never the case in previous appearance of the term in your manuscript.

Line 221, I believe that you wanted to say ‘all’ not ‘tall’; same line, I don’t get the use of the adjective ‘short’

Line 307, I believe that you wanted to say ‘went’ not ‘vent’

Discussions

This paragraph is very long. I suggest divisions in sub-paragraphs based on cores of subject

Reviewer 2 Report

Comments and Suggestions for Authors

Dear authors,

Many thanks for your paper, I appreciated very much to read it, especially as it is one of the few paper that analyze this kind of problems in terms of qualitative analysis.

Below some consideration that I think could help you to improve your paper.

In introduction section, authors refers to “digital native”, really, some years ago we talk about digital native with reference to a specific court of people, today I think it is important to be more specific as technology is so fast, so, I recommend to  apply a distinction between “pure digital native” (0-12), “millennials” (13-18), “Spure digital native”(19-25) as reported on Wired

Considering that authors refer to CIV (cyber interpersonal violence), it is important to define it right away. The definition appear only in line 52, too later…a reader imagine some things that could be not reflect the correct definition of CIV.

Line 55-56 “..Wright noted a great deal of variation in 55 cyber aggressive actions and distinguished two distinct kinds” , last words appear in red, but what are this two kinds? In line 57-61 there is only the second kind. Please re-write from line 56.

Line 96-97 Facebook probably in 2019 was a famous SNS in Portugal, but I think that today (2024) it isn’t so, please actualize the information

Line 147 elementary? better primary schools?

In methodology section there are terms that could lead reader in confusion. For example, you refers to interviewer (line 158), but as it is a focus group the people involved are: conductor and observer.

Procedure section need to be better organized and explained. For a better reading perhaps the order of the paragraphs could be changed. As the procedure section is after participant section, I'm expect to found quickly how participant was selected, starting from agreement with headmaster, parents etc…Please try to reposition the parts from the line 151 to 180 for a more fluid and linear reading.

Please provide an explication regarding the chosen criteria, why “some of them only had boys 181 or girls from several grades, and other had participants from the same grade, all picked 182 randomly.”  (line 181-182)?

As discussion was audio recorded you have to mention if participant was agreeing to recording and how and when did you ask it.

Please provide the semi-structured interview, it will be insert in the paper or as file added. If you are in restriction by the project yet, you can insert the macro-areas of discussion.

I believe that the references are wrote not in line with the journal recommendation (i.e. doi is missing, italics etc.), please proofread. 
